# miR-183/96/182 Cluster Regulates the Development of Bovine Myoblasts through Targeting *FoxO1*

**DOI:** 10.3390/ani12202799

**Published:** 2022-10-17

**Authors:** Wenxiu Ru, Kunpeng Liu, Jiameng Yang, Jianyong Liu, Xinglei Qi, Bizhi Huang, Hong Chen

**Affiliations:** 1Key Laboratory of Animal Genetics, Breeding and Reproduction of Shaanxi Province, College of Animal Science and Technology, Northwest A&F University, Xianyang 712100, China; 2Yunnan Academy of Grassland and Animal Science, Kunming 650212, China; 3Bureau of Animal Husbandry of Biyang County, Zhumadian 463700, China

**Keywords:** bovine myoblasts, miR-183/96/182 cluster, *FoxO1*, proliferation, differentiation

## Abstract

**Simple Summary:**

In this work, we identified that the miR-183/96/182 cluster was highly expressed in bovine embryonic muscle; meanwhile, it widely existed in other organizations. Functional assays indicated that the miR-183/96/182 cluster targets the *FoxO1* gene to regulate the proliferation and differentiation of bovine myoblasts.

**Abstract:**

Muscle development is an important factor affecting meat yield and quality and is coordinated by a variety of the myogenic genes and signaling pathways. Recent studies reported that miRNA, a class of highly conserved small noncoding RNA, is actively involved in regulating muscle development, but many miRNAs still need to be further explored. Here, we identified that the miR-183/96/182 cluster exhibited higher expression in bovine embryonic muscle; meanwhile, it widely existed in other organizations. Functionally, the results of the RT-qPCR, EdU, CCK8 and immunofluorescence assays demonstrated that the miR-183/96/182 cluster promoted proliferation and differentiation of bovine myoblast. Next, we found that the miR-183/96/182 cluster targeted *FoxO1* and restrained its expression. Meanwhile, the expression of *FoxO1* had a negative correlation with the expression of the miR-183/96/182 cluster during myoblast differentiation. In a word, our findings indicated that the miR-183/96/182 cluster serves as a positive regulator in the proliferation and differentiation of bovine myoblasts through suppressing the expression of *FoxO1*.

## 1. Introduction

Skeletal muscle is the most abundant tissue accounting for most of the body weight and participates in movement and metabolism [1]. For livestock, skeletal muscle development is an important factor affecting meat yield and quality. Skeletal muscle originates from mesenchymal stem cells (MSC) in the embryonic mesoderm, which could change to myogenic progenitor cells. The myogenic progenitor cells differentiate into mononuclear myoblasts; then, the myoblasts further experience proliferation, differentiation and fusion into multinuclear myotubes, which ultimately form muscle fibers [2]. Once the mature muscle has formed, myogenic progenitor cells will enter quiescence and exist as muscle satellite cells, which could participates in the repair of muscle fibers [3]. It is now generally accepted that this complex and long-term process is precisely coordinated by the myogenic regulatory factors (MRFs), containing Myogenic differentiation 1 (*MyoD1*), Myogenic regulatory factor 4 (*Mrf4*), Myogenin (*MyoG*) and Myogenic factor 5 (*Myf5*) [4]. Tuning these genes’ expression in muscle development is realized through a transcriptional and post-transcriptional network. Increasing evidence has suggested that noncoding RNAs (ncRNAs) involving post-transcriptional regulation is a vital factor in muscle development, including long noncoding RNA [5], circular RNA and small RNA [6,7]. Although the genetic and molecular pathways of regulating muscle development have been well-established in the past decades, numerous unknown regulatory molecules and mechanisms involved in this process remain unidentified.

MicroRNAs (miRNA), a class of ~22 nucleotide small noncoding RNAs, are highly conserved and do not possess potential coding. The seed region of miRNA located in 2~8 nucleotides at the 5′ end normally conjugates with the 3′-untranslated region (3′ UTR) of the target mRNA. By this means, miRNAs are capable of decaying or impeding protein translations of the target mRNAs [8]. Numerous studies have underlined the key roles of miRNAs in regulating skeletal muscle development through their inhibiting effects on several myogenic regulatory factors and important signaling pathways. The MyomiR family, a class of miRNA specifically expressed in muscles, includes miR-133, miR-1, miR-208a/b, miR-206, miR-486 and miR-499 [9]. For instance, miR-206, which is highly and exclusively expressed in muscles, was positively regulated by *MyoD* and targeted *Pax7* to promote terminal differentiation [10], while some non-muscle-specific miRNAs also participated in regulating muscle development, such as miR-24, miR-27a, miR-125b, miR-29, miR-486, miR-221/222 and miR-214 [11]. Moreover, the indispensable functions of miRNAs in muscle development have verified that the deficiency of Dicer in skeletal muscle reduced the production of muscle miRNAs and led to embryonic death during embryogenesis [12].

The conserved miR-183/96/182 cluster is one of the most studied miRNA clusters, which is situated at a 5-kb genomic region in humans and mice and possesses similar seed sequences [13]. Many upstream regulators, including *Wnt/beta-Catenin*, the *p21-ZEB1* complex, *GSK3β*, *MyoD* and so on, were certified to regulate the expression of the miR-183/96/182 cluster [14,15,16,17]. Increasing studies have shown that the miR-183/96/182 cluster plays an important role in tumorigenesis, cancer progression, tumor invasion and metastasis. Notably, an inconsistency has arisen with respect to the functions of the miR-183/96/182 cluster in various tumor cells [18], which prompted that its versatility in different biological processes. Nevertheless, there are research gaps in the regulatory role of the miR-183/96/182 cluster in muscle development, especially in livestock animals. Therefore, we attempted to explore the function and mechanism of the miR-183/96/182 cluster in bovine skeletal muscle development. In this study, we found that the miR-183/96/182 cluster accelerated the proliferation and differentiation of bovine myoblasts by targeting the *FoxO1* gene, which enriched the network of miRNAs regulating bovine muscle development.

## 2. Materials and Methods

### 2.1. Sample Preparation

Muscle samples in Qinchuan cattle were collected at the adult stage (24 months) and embryonic stage (90 days) from a local livestock farm in Xi’an (Shannxi, China). Other tissue samples included a liver, kidney, heart, lung and spleen that were obtained at the embryonic stage (90 days), and each sample was collected from three cattle about the same age. After surgical removal, all samples were placed in liquid nitrogen to snap-freeze and then kept at −80 °C until RNA isolation. Our study protocols were approved by the Animal the Ethics Committee of Northwest A&F University.

### 2.2. Cell Culture

As previously described, the primary myoblasts of bovines were obtained from the longissimus muscle at the fetal stage (90 days) [19]. Briefly, the stripped muscle tissues were cut into pieces, then digested with collagenase I in 37 °C for 2 h, whereafter the digested muscles were filtered using a 200-mesh filter, and the filtrate was washed with PBS and centrifuged at 1500 rpm three times. Finally, the collected cells were cultured in DMEM (BI, ISR) supplemented with 20% fetal bovine serum (BI, ISR) and 2% penicillin–streptomycin (Biosharp, Anhui, China). When 90% confluence was reached, the medium was changed to DMEM with 2% horse serum and 2% penicillin–streptomycin to induce cell differentiation. The HEK-293T cells were cultured in DMEM with 10% FBS and 1% penicillin–streptomycin. All cells were cultured in a humidified incubator at 37 °C with 5% CO_2_.

### 2.3. Vector Construction and Transfection

The fragment of bovine *FoxO1* 3′UTR containing the miR-183/96/182 cluster wild binding site or mutant sites was cloned into the psiCHECK2 vector. These vectors were verified by sequencing. The mimics and inhibitors of the miR-183/96/182 cluster were synthesized by General Biol (Chuzhou, China) to the overexpression or knockdown of the miR-183/96/182 cluster, respectively. All vectors and mimics or inhibitors were transfected into myoblasts using Troubfect (Thermo Fisher Scientific, Waltham, MA, USA).

### 2.4. RNA Extraction and RT-qPCR Analysis

The total RNA from tissues and cells was extracted using the Trizol reagent (AG, Beijing, China) and was reverse transcribed using the RT reagent kit (AG, Beijing, China). For mRNA, random and oligo (dT) primers were used to synthesize cDNA, and for miRNA, random and stem-loop primers were used to synthesize cDNA. The RT-qPCR assay was performed using the SYBR Green Kit (Vazyme, Nanjing, China) on the CFX96 System (Bio-Rad, Hercules, CA, USA). We applied *β-actin* and *U6* as an internal control for the mRNA and miRNA. The quantitation data was analyzed by the 2^−ΔΔCt^ method, and each sample was replicated three times. All primers are listed in Appendix A.

### 2.5. Cell Proliferation Assay

The myoblasts were cultured in 96-well plates and then transfected after the cell density reached 70–80%. For the CCK-8 assay, each well was treated with 10 μL of CCK-8 (UE, Suzhou, China) and incubated for 2 h at 37 °C in the dark. The absorbance of each sample was detected using a microplate reader at 450 nm, and each sample was replicated eight times. The EdU Cell Proliferation Kit (Beyotime, Shanghai, China) was used to measure the capacity of cell proliferation, and Hoechst 33342 was used to stain the nuclei. Each sample was replicated three times. Finally, the images were observed by fluorescence microscopy (AMG EVOS, SEA, USA).

### 2.6. Immunofluorescence Assay

After transfection, bovine myoblast differentiation was induced for 4 d. Then, the cells were washed with PBS and fixed with 4% paraformaldehyde for 30 min. After washing, the cells were permeabilized with 0.5% Triton X-100 for 15 min and blocked with 5% BSA for 30 min, following incubation at 4 °C overnight with antibody-*MyHC* diluted 1:250 (GeneTex, Irvine, CA, USA). Next, we used the homologous fluorescent secondary antibody (Immunoway, Plano, TX, USA) diluted 1:500 to incubate cells for 2 h at room temperature. Hoechst 33342 was used to stain the nuclei. Finally, the images were observed under a fluorescent microscope.

### 2.7. Luciferase Reporter Assay

The psiCHECK2-FoxO1-WT or psiCHECK2-FoxO1-Mut and miR-183/96/182 cluster mimics were co-transfected into HEK293T cells using Troubfect (Thermo Fisher Scientific, Waltham, MA, USA). After transfection for 24 h, the cells were lysed, and next, we used the Dual-Luciferase Reporter Assay Kit (Promega, MDN, USA) to detect luciferase activities according to the manufacturer’s instructions. Finally, the ratios of renilla and firefly activity were calculated, and each sample was replicated eight times.

### 2.8. Statistical Analysis

We used GraphPad Prism 8.0 (GraphPad, San Diego, CA, USA) and SPSS 22.0 (SPSS, Chicago, IL, USA) to analyze the data. Significance analyses between two groups were performed by an independent sample *t*-test, and for three or more groups, one-way analysis of variance (ANOVA) was used to compare any discrepancies. All data were presented as the mean ± SEM. A statistical significance was indicated as * *p* < 0.05, ** *p* < 0.01, *** *p* < 0.001 or **** *p* < 0.0001.

## 3. Results

### 3.1. The Expression Characteristics of the miR-183/96/182 Cluster in Cattle Tissues

There have been numerous studies that have proven the important roles of the miR-183/96/182 cluster in cell proliferation, apoptosis and differentiation in various cancer cells [18]. In this study, we aimed to explore whether the miR-183/96/182 cluster also has regulatory functions in bovine myoblasts. Firstly, we downloaded the mature sequences of miR-183, miR-96 and miR-182 to examine their conservatism in different species. The results demonstrated that the miR-183/96/182 cluster is highly conserved among species (Figure 1A). Through quantifying the expression features of the miR-183/96/182 cluster, we found that it exhibits varying expression patterns in various tissues in bovines (Figure 1B). In addition, the miR-183/96/182 cluster had a significantly higher presence in embryonic muscle (Figure 1C), which suggested that it may be a positive regulator in embryonic muscle development.

### 3.2. miR-183/96/182 Cluster Promoted Bovine Myoblast Proliferation

The miR-183/96/182 cluster has been shown to promote most cancer cell proliferation [18]. To verify whether the miR-183/96/182 cluster influences bovine myoblast proliferation, we transfected bovine myoblasts with miR-183/96/182 mimics to increase its expression. The expression level of miR-183/96/182 was significantly higher than the control group (Figure 2A). A RT-qPCR assay was used to detect the expression of marker genes of proliferation. Our results demonstrated that the overexpression of the miR-183/96/182 cluster remarkably enhanced the expression of *PCNA*, *CDK2* and *cyclin D*; meanwhile, the level of *P21* was significantly reduced (Figure 2B). The EdU proliferation assays revealed that the overexpression of miR-183/96/182 significantly increased the number of EdU-positive cells, and silencing miR-183/96/182 reduced the positive cells (Figure 2C,D). In addition, the results of the CCK8 assays showed that overexpressing miR-183/96/182 could increase the vitality of myoblasts, and silencing miR-183/96/182 could inhibit the vitality of the myoblasts (Figure 2E). Thus, our results showed that the miR-183/96/182 cluster can facilitate the proliferation of bovine myoblasts.

### 3.3. miR-183/96/182 Cluster Promotedpromote Bovine Myoblast Differentiation

Next, we verified the potential functions of the miR-183/96/182 cluster in myoblast differentiation. After the transfection of miR-183/96/182 mimics for 24 h, we used 2% horse serum medium to induce myoblast differentiation until day 3. Subsequently, we detected the expression level of the differentiation marker genes by RT-qPCR. As shown in Figure 3A, there is a significant rise in the expression level of the differentiation marker genes, including *MyoG*, *MyoD* and *Myf5*. However, silencing the miR-183/96/182 cluster reduced the expression of differentiation marker genes at the mRNA level. Additionally, the immunofluorescence assay also demonstrated that overexpressing the miR-183/96/182 cluster could prompt the myotube to become bigger (Figure 3B). Inversely, the myotube became lesser after silencing the miR-183/96/182 cluster (Figure 3B). Consequently, all the results suggested that the miR-183/96/182 cluster promoted bovine myoblast differentiation.

### 3.4. FoxO1 as a Target Gene of the miR-183-96-182 Cluster

*FoxO1*, a well-known direct target of the miR-183/96/182 cluster, has been confirmed in various types of cancers, such as prostate [20], liver [16], breast [21], lymphoma [22] and so on. In addition, *FoxO1* has also been proven to participate in muscle differentiation and glucose and lipid metabolism in skeletal muscle [23,24,25]. However, whether the miR-183/96/182 cluster regulates *FoxO1* in bovine muscle is still unclear. As expected, we found that *FoxO1* was potentially targeted by the miR-183/96/182 cluster, and the distribution of potential binding sites is demonstrated in Figure 4A. To investigating the binding of the miR-183/96/182 cluster and *FoxO1*, we constructed a dual-luciferase reporter system of *FoxO1* 3′UTR (wild type) (Figure 4B). The results indicated that the luciferase activity of pCK-FoxO1-WT was notably suppressed in HEK293T cells after co-transfection with the miR-183/96/182 cluster mimics (Figure 4C). Analogously, we further used the vector of pCK-FoxO1-MUT to verify this interaction (Figure 4A,D). There was no longer a response in the pCK-FoxO1-MUT system when the miR-183/96/182 cluster mimics were transfected (Figure 4D). The RT-qPCR assay indicated that the expression of *FoxO1* was markedly decreased after overexpressing the miR-183/96/182 cluster in bovine myoblasts (Figure 4E). Additionally, we found that the expression of *FoxO1* had a negative correlation with the expression of the miR-183/96/182 cluster during myoblast differentiation (Figure 4F). Taken together, these results demonstrated that the miR-183/96/182 cluster regulates the proliferation and differentiation of bovine myoblasts by targeting *FoxO1*.

## 4. Discussion

Muscle development is precisely coordinated by members of the myocyte enhancer factor 2 (*MEF2*) family and myogenic regulatory factors (MRFs) [26,27]. In recent years, miRNA have been certified to mediate the post-transcriptional regulation of gene expression by RNA interference [13]. To date, numerous studies have established a powerful role of miRNAs in cell differentiation, growth, apoptosis and development, as we all know that myoblast proliferation and differentiation are the key factors affecting muscle development. Therefore, the potential effects of miRNAs in myoblast proliferation and differentiation cannot be ignored. As we all know, the conserved miR-183/96/182 cluster is one of the most studied miRNA clusters, which possesses similar seed sequences to target identical genes; meanwhile, it plays a crucial role by coordinating the key genes in various cellular processes [13]. Consistent with our analysis, the mature fragment of the miR-183/96/182 cluster is highly conserved in different species (human, mouse, bovine, rat, pig and chicken). However, it is still unclear whether the miR-183/96/182 cluster has a regulatory function in bovine myoblasts. In this study, we indicated that the miR-183/96/182 cluster serves as a positive regulator in the proliferation and differentiation of bovine myoblasts through suppressing the expression of *FoxO1*.

Previous studies have demonstrated that the miR-183/96/182 cluster possessed an accelerated role in cell proliferation in most types of cancer. A classic example is that miR-182 and miR-183 facilitate cell proliferation and tumor invasion by inhibiting *PDCD4* in various cancer cells, which is a typical tumor suppressor gene [28,29]. Interestingly, miR-96 was found to weaken pancreatic cancer cell proliferation, and miR-183 suppressed gastric cancer proliferation in the past few years [30,31]. These discrepant results may be caused by different types of cancer cells or competition between target genes or the involvement of different signaling pathways. Therefore, it is necessary for us to continue to explore the regulatory functions of the miR-183/96/182 cluster in various cellular environments and different life processes. Recent studies have reported the modulating capability of miR-96 and miR-183 in muscle oxidative, and the results showed that miR-96 and miR-183 can inhibit glucose utilization and fat catabolism [32].

Here, we further exhibited that the miR-183/96/182 cluster plays a vital role in promoting skeletal muscle proliferation and differentiation. The miR-183/96/182 cluster exhibited widely different expression patterns across bovine tissues, which is also consistent with its extensive functions in various tissues and cell development. Specifically, we observed a higher level of the miR-183/96/182 cluster in embryonic muscle tissue than adult muscle. The embryonic period is a critical stage for muscle growth and development; in addition, the miR-183/96/182 cluster can accelerate the proliferation and differentiation of bovine myoblasts, suggesting that miR-183/96/182 plays a crucial part in myogenesis. Notably, several studies in mice have confirmed that miR-96-5p and miR-183-5p, via suppressing *FHL1*, impede the differentiation and fusion of myoblasts [33,34]. Contrary to their results, we found that the miR-183/96/182 cluster promoted the proliferation and differentiation of bovine myoblasts through using RT-qPCR, EdU, CCK8 and immunofluorescence assays. This positive regulatory effect of the miR-183/96/182 cluster for bovine myoblasts is essential for skeletal muscle development. To be noted, the differences between species are likely mainly responsible for the function differences. Additionally, there may be competition between target genes in a specific context.

Mechanism studies have shown that *FoxO1* acts as a target gene of the miR-183/96/182 cluster to mediate bovine myoblast development. It is known that *FoxO1*, a member of the “O” subclass of the (FOX) family, has a considerable effect on various cellular physiological process. *FoxO1* is found in most muscle types and plays a vital regulation in myoblast proliferation and differentiation, muscle growth and metabolism [35]. The function of *FoxO1* in muscle differentiation has been widely reported. Some researchers have demonstrated that *FoxO1* facilitates myotube fusion in mouse primary myoblasts [36]. However, other studies have revealed that *FoxO1* is an inhibitor for muscle differentiation. At an early myogenesis stage, *FoxO1* has been reported to impinge on the nutrient-sensing mTOR pathway, the Notch pathway and myostatin to repress myoblast differentiation [37,38,39]. The study in vivo showed that *FoxO1* transgenic mice exhibited a remarkable reduction in muscle mass and the decreased expression of type I fiber genes and impaired skeletal muscle production [25]. Generally, most studies support *FoxO1* as an inhibitor in myogenesis. Furthermore, increasing studies have verified that *FoxO1* is a direct target gene to the miR-183-96-182 cluster. In bovine ovaries, the miR-183-96-182 cluster promotes granulosa cell proliferation, cycle progression and restrains apoptosis through targeting *FoxO1* [40,41]. In the present study, we also confirmed that *FoxO1* was the target gene of the miR-183-96-182 cluster using the dual-luciferase reporter assay, and overexpressing miR-183/96/182 cluster led to decreasing levels of *FoxO1* mRNA in bovine myoblasts. Importantly, we observed that the expression of *FoxO1* was negatively correlated with the expression of the miR-183/96/182 cluster during myoblast differentiation. Thus, our results established a pattern that the miR-183/96/182 cluster regulates the proliferation and differentiation of bovine myoblasts by targeting *FoxO1*. It is worth noting that we just uncovered a potential mechanism that the miR-183/96/182 cluster regulates muscle development, but there are other target genes of the miR-183/96/182 cluster that unceasingly need to be elucidated.

## 5. Conclusions

Overall, we characterized the miR-183/96/182 cluster in bovine muscle and identified the miR-183/96/182 cluster as a positive regulator in the differentiation and proliferation of bovine myoblasts. Mechanistically, we found that the miR-183/96/182 cluster targets the *FoxO1* gene to regulate the proliferation and differentiation of bovine myoblasts. Our findings not only confirmed the universality of the regulatory functions of the miR-183/96/182 cluster in various biochemical processes but also provided a theoretical basis to clarify skeletal muscle development in bovines from a layer of noncoding RNAs.

## Figures and Tables

**Figure 1 animals-12-02799-f001:**
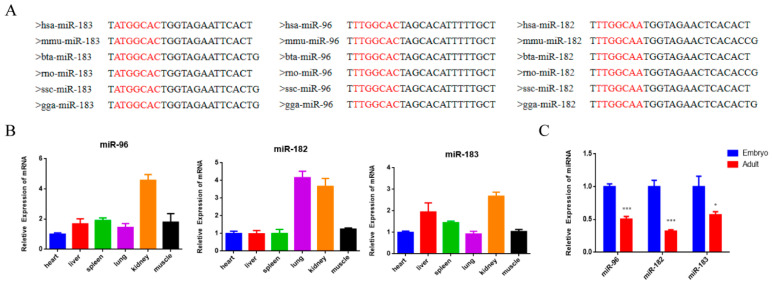
The expression characteristics of the miR-183/96/182 cluster in cattle. (**A**) Conservative analysis of the mature sequence of the miR-183/96/182cluster among different species. The red letters show the miRNA seed sequence. (**B**) The expression feature of the miR-183-96-182 cluster in different tissues of fetal cattle. (**C**) The expression level of the miR-183/96/182 cluster in fetal and adult bovine muscle tissues. Data are presented as the means ± SEM. * *p* < 0.05 and *** *p* < 0.001.

**Figure 2 animals-12-02799-f002:**
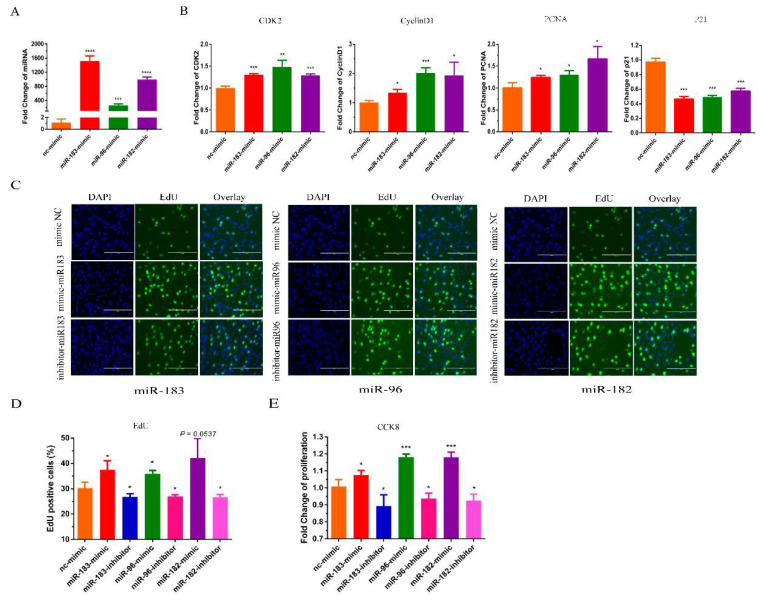
The miR-183/96/182 cluster promotes bovine myoblast proliferation. (**A**) The detection of expression efficiency after the transfection of miR-183/96/182 cluster mimics. (**B**) The expression levels of the cell proliferation genes *CyclinD1*, *PCNA, CDK2* and *P21* mRNA were detected by RT-qPCR. (**C**,**D**) EdU-positive cells were detected after transfection of the miR-183/96/182 cluster mimics and inhibitors. (**E**) Proliferation in bovine myoblasts was performed by the CCK-8 assay. Data are presented as the means ± SEM. * *p* < 0.05, ** *p* < 0.01, *** *p* < 0.001 and **** *p* < 0.0001.

**Figure 3 animals-12-02799-f003:**
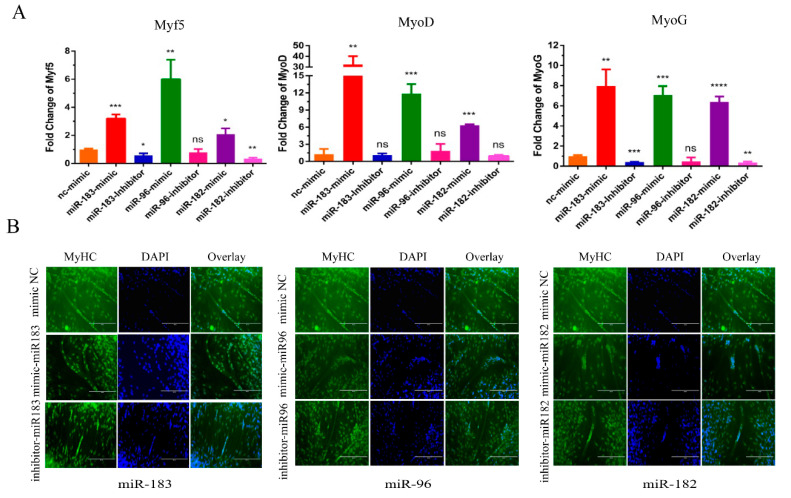
The miR-183/96/182 cluster promotes bovine myoblast differentiation. (**A**) The expression levels of cell differentiation genes *MyoG*, *MyoD* and *Myf5* were detected by RT-qPCR. (**B**) Immunofluorescence (*MyHC*) was performed to evaluate the cell differentiation. Data are presented as the means ± SEM. NS represented no difference. * *p* < 0.05, ** *p* < 0.01, *** *p* < 0.001 and **** *p* < 0.0001.

**Figure 4 animals-12-02799-f004:**
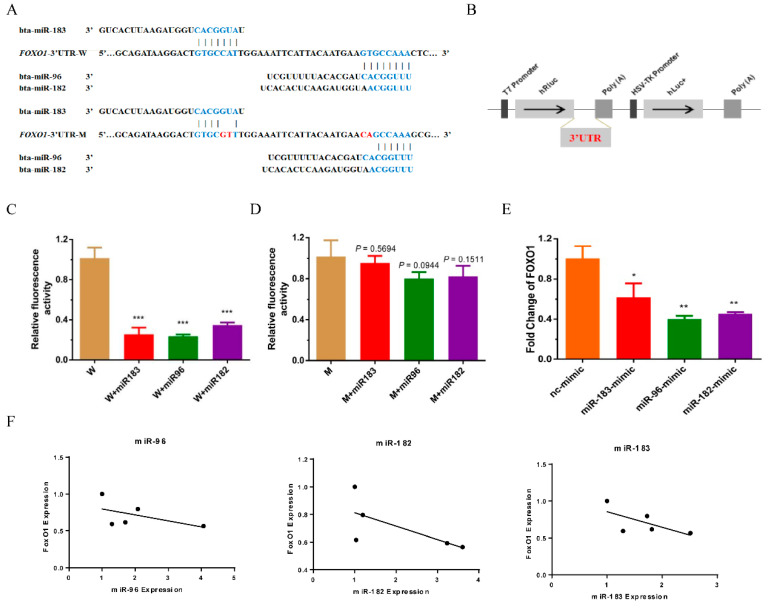
*FoxO1* was the target of the miR-183/96/182 cluster. (**A**) The distribution of potential binding sites of the miR-183/96/182 cluster in the 3′UTR of *FoxO1*. The red letters indicate the certain mutated bases. (**B**) The illustration shows the construct of a dual-luciferase reporter system. (**C**,**D**) The luciferase activity of the *FoxO1* (wild or mutant) psi-check2 reporter vector was detected after co-transfection of the miR-183/96/182 cluster mimics in HEK293T cells. (**E**) The mRNA of *FoxO1* were detected by RT-qPCR after transfection of the miR-183/96/182 cluster mimics and inhibitors. (**F**) The correlation analysis on the expression of *FoxO1* and the miR-183/96/182 cluster during myoblast differentiation. Data are presented as the means ± SEM. * *p* < 0.05, ** *p* < 0.01 and *** *p* < 0.001.

## Data Availability

All data generated or analyzed during this study are included in this published article.

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
