# Peer review of "miR-183/96/182 Cluster Regulates the Development of Bovine Myoblasts through Targeting FoxO1"

_animals, 2022, doi:10.3390/ani12202799_

Round 1

Reviewer 1 Report

The manuscript is very interesting and very well concerned. I have only two major concerns: 1) English must be improved. Some parts of the text are very difficult to understand 2) The discussion is poorly written. The authors should develop their discussion because they have such great results.

Methodology and statistical analysis seem to be correct

Author Response

1. English must be improved. Some parts of the text are very difficult to understand.
Response: We are very sorry because our manuscript has some minor grammar errors. We tried our best to improve the English writing and made some changes in the manuscript and the modifications in the article are marked in red.

2. The discussion is poorly written. The authors should develop their discussion because they have such great results.

Response: Thank you very much for your advice. We have made changes in the discussion and the modifications in the article are marked in red. We tried our best to develop our results and I hope to get your approval.

Reviewer 2 Report

The manuscript presents the roles of miR-183/96/182 cluster in development of bovine myoblast and identified Foxo1 was the target of the miR-183/96/182 cluster. The data sound great and the conclusion is of significance for muscle development researches. The following points are suggested to be addressed further.   1. Line 95: How to assure the purity of bovine myoblasts?

2. Fig 4: Foxo1 expression should also be analyzed during myoblast development and negatively correlated with miR-183/96/182 cluster expression.

3. The writing of this manuscript is edited well totally, but some mistakes could be still found.

In line 48: " Increasing evidence has " should be " Increasing evidences have ". 

In line 64: " targets " should be " targeted". 

In line 158: " We aim…" should be " We aimed… ". 

In line 183: " promote" should be " promotes… ". 

Please check the whole MS and revise these mistakes.  

Author Response

1. Line 95: How to assure the purity of bovine myoblasts?

Response: Myoblasts were obtained from longissimus muscle of fetal cattle using the procedure for isolation of primary cultures in mice (Rando and Blau 1994; Conboy and Rando 2002). The method of myoblast separation in this study has been widely used (Miyake et al., 2012). The paired box transcription factor Pax7 is a key regulator of skeletal muscle stem cells and is required along with Pax3 to generate skeletal muscle precursors. Therefore, we can make sure the presence of myoblasts by fluorescence labeling Pax7 and Pax3. Our team have stained with anti-PAX7 fluorescent antibody, but the data have not been published. And in this study, the myotubes could be labeled with anti-MyHC fluorescent antibody after 4 days of differentiation. In addition, we also used differential adhesion method to eliminate the fibroblasts. So, we can obtain bovine myoblasts.

2. Fig 4: Foxo1 expression should also be analyzed during myoblast development and negatively correlated with miR-183/96/182 cluster expression.

Response: Thank you very much for your advice. We analyzed the expression patterns of FOXO1 and miR-183/96/182 cluster at 0, 1, 2, 3, 4 days of myoblast differentiation. And the results showed that the FOXO1 expression is negatively associated with miR-183/96/182 cluster expression during myoblast differentiation (Fig 4F). The corresponding explanation are also added in line 215-217 and marked in red.

3. The writing of this manuscript is edited well totally, but some mistakes could be still found.

Response: We are very sorry because our manuscript has some minor grammar errors. We have corrected the error in our manuscript as you requested and examined the manuscript carefully. Some changes in the manuscript and the modifications in the article are marked in red.

Reviewer 3 Report

Comments for authors

The article deals with the important issue related to muscle development which is an essential factor affecting meat yield and quality.

The manuscript is clear, relevant for the field and presented in a structured manner but professional language correction is needed.

A detailed list of notes on my script is provided below:

Abstract:

1.      Row 16 - the word „Muscle” should not be bold

Introduction:

1.      Row 34 – instead of „whose” should be „which”

2.      Row 62 – add a dot after the abbreviation „etc” – etc.

3.      Row 77 – here is a grammatical mistake, should be „found” instead of „founded”

Materials and Methods:

Sample preparation

1.      Row 83 – instead of „of” should be „in”

2.      There is not enough information on collecting and preparing samples here. It was not stated how many samples were taken and from how many animals. Moreover, there is no information on how these samples were stored, i.e. whether they were suspended in any solution etc.

Cell Culture

1.      Row 91 – Remove „a” before 1500 and „in” before three

RNA extraction and qRT-PCR analysis

1.      Row 105 – remove dot after parentheses

2.      Row 110 – add space after „applied”

3.      Row 111 – author mentioned Table S1 but it is not in the manuscript

Results

1.      Graphs are unreadable

2.      Row 201 – the word „Cluster” should start with a lowercase letter

3.      Row 202 - here is a grammatical mistake, should be „has” instead of „have”

References

1.      The references chapter needs to be improved. The list of publications has not been prepared in accordance with the journal's requirements

Author Response

1. Professional language correction is needed. A detailed list of notes on my script is provided below.

Response: Thank you very much for your advice. We are very sorry because our manuscript has so many minor grammar errors. Therefore, we have made correction according to the Reviewer’s comments and the modifications in the article are marked in red. Furthermore, we examined the manuscript carefully and our best to improve the language.

2. There is not enough information on collecting and preparing samples here.  It was not stated how many samples were taken and from how many animals.  Moreover, there is no information on how these samples were stored, i.e. whether they were suspended in any solution etc.

Response: Thank you very much for your advice. We have listed the tissue samples and the number of cattle as you suggested. Also, we have supplemented the method of samples storage. The corresponding explanation are added in line 90-93 and marked in red.

3. Row 111 – author mentioned Table S1 but it is not in the manuscript.

Response: We are very sorry for the trouble caused. The table S1 is presented in the supplementary information. We will submit “supplementary information” later.

Round 2

Reviewer 3 Report

Przyjmuję rękopis w obecnej formie